# Instance-Level Generation for Representation Learning

**Yankun Wu**                                                             *yankun@is.ids.osaka-u.ac.jp*
*The University of Osaka*

**Zakaria Laskar**                                                 *zakaria.laskar@iisertvm.ac.in*
*School of Data Science, IISER Thiruvananthapuram*

**Giorgos Kordopatis-Zilos**                                              *kordogeo@fel.cvut.cz*
*VRG, FEE, Czech Technical University in Prague*

**Noa Garcia**                                                        *noagarcia@ids.osaka-u.ac.jp*
*The University of Osaka*

**Giorgos Tolias**                                                         *toliageo@fel.cvut.cz*
*VRG, FEE, Czech Technical University in Prague*

**Reviewed on OpenReview:** *https://openreview.net/forum?id=T3JgJXH3ZK*

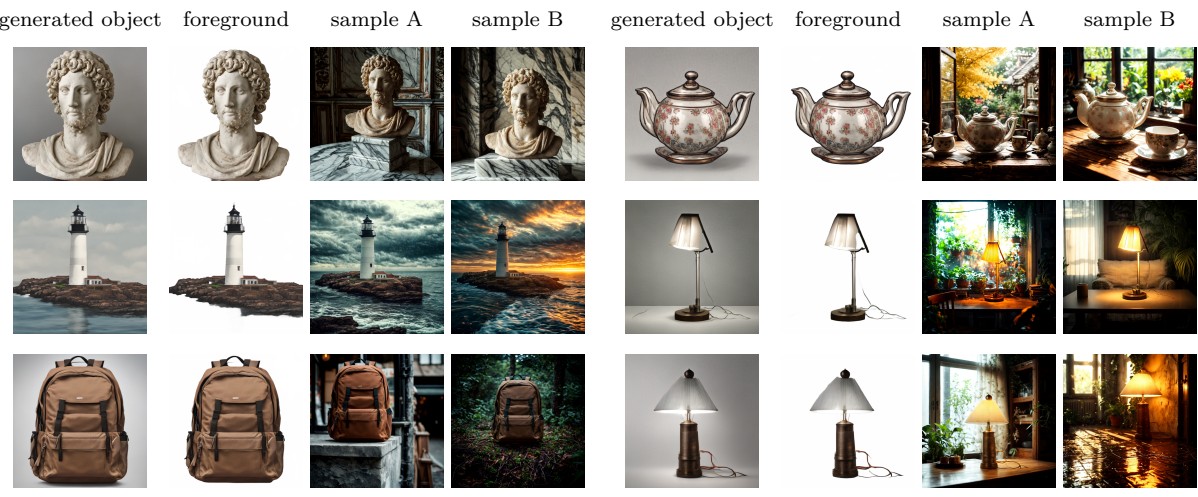

Figure 1: Examples of images generated for learning instance-level representations. Given an object generated by a generative diffusion model (column 1), the foreground is segmented (column 2) and different background variations are added (columns 3 & 4), producing images of the same instance under diverse conditions.

## Abstract

Instance-level recognition (ILR) focuses on identifying individual objects rather than broad categories, offering the highest granularity in image classification. However, this fine-grained nature makes creating large-scale annotated datasets challenging, limiting ILR's real-world applicability across domains. To overcome this, we introduce a novel approach that synthetically generates diverse object instances from multiple domains under varied conditions and backgrounds, forming a large-scale training set. Unlike prior work on automatic data synthesis, our method is the first to address ILR-specific challenges without relying on any real images. Fine-tuning foundation vision models on the generated data significantly improves retrieval performance across seven ILR benchmarks spanning multiple domains. Our approach offers a new, efficient, and effective alternative to extensive data collection and curation, introducing a new ILR paradigm where the only input is the names of the target domains, unlocking a wide range of real-world applications. The code and pretrained models are publicly available at https://github.com/yankungou/ILGen.

# 1 Introduction

Object recognition and retrieval span multiple levels of granularity, from semantic-level labels (Russakovsky et al., 2015) to fine-grained categories (Gosselin et al., 2014; Krause et al., 2015), and the most detailed form, i.e. instance-level recognition (ILR) (Ypsilantis et al., 2021). Unlike semantic recognition, which groups objects into broad classes, ILR identifies unique object instances, treating each real-world entity as its own category. This extreme granularity makes ILR particularly challenging.

ILR has applications in domains such as landmarks (Weyand et al., 2020; Philbin et al., 2007; 2008), artwork (Ypsilantis et al., 2021), products (Oh Song et al., 2016; Peng et al., 2020), fashion (Liu et al., 2016), and everyday objects (Wang & Jiang, 2015; Kordopatis-Zilos et al., 2025). However, large-scale training data remains a major bottleneck. Unlike semantic or fine-grained recognition, where class names help structure data and reduce false negatives, ILR requires exhaustive, instance-specific annotations, an expensive and labor-intensive process. Single-domain datasets rely on manually curated ground truth, while multi-domain datasets often lack dedicated training sets (Wang & Jiang, 2015; Kordopatis-Zilos et al., 2025). Collecting images of the same instance under different conditions further compounds the challenge, slowing progress.

To address this, we propose a novel pipeline that automatically generates images of unique objects under diverse conditions, enabling instance-level representation learning without manual data collection. The pipeline requires only the name of one or more domains, e.g. "everyday objects" or "artworks", as input and outputs a representation model fine-tuned for those domains. A large language model (LLM) (Hurst et al., 2024) generates a list of relevant object categories, and a generative diffusion model (GDM) (Sauer et al., 2024; Rombach et al., 2022) synthesizes images for those categories. We assume that generations from a given seed define an instance-level class, while different seeds correspond to distinct classes, and validate this assumption through experimental analysis and user study. To ensure diversity, we introduce background and lighting variations using ICLight (Zhang et al., 2025).

The generated instances (see Figure 1) are used to fine-tune a foundational vision encoder such as SigLIP (Zhai et al., 2023). We adopt a metric learning approach (Patel et al., 2022), treating images of the same instance as positives and others as negatives, and optimize an information retrieval metric across large batches. The resulting representation improves over the base model across multiple ILR benchmarks, including artwork, landmark, and product datasets.

This is the first work to learn a single representation model that generalizes across diverse ILR domains while providing an effective alternative and complement to large-scale real data. While prior research explored synthetic training data (Peng et al., 2015; Fan et al., 2024; Tian et al., 2024), our method is the first tailored specifically for ILR. The pipeline synergistically integrates LLMs and GDMs, leveraging rapid advances in both fields and remaining adaptable to future improvements.

# 2 Related work

**Instance-level representations** Instance-level recognition requires image representations that capture fine-grained object details while distinguishing them from numerous semantically similar classes. Generic models like ResNet (He et al., 2016) and CLIP (Radford et al., 2021) struggle in this setting, as they prioritize high-level semantics over instance-specific features. A common solution is fine-tuning pre-trained backbones on domain-specific datasets—such as artwork (Ypsilantis et al., 2021), landmarks (Lee et al., 2022; Shao et al., 2023; Cao et al., 2020; Suma et al., 2024), or products (Patel et al., 2022; Ramzi et al., 2022)—to enhance their ability to differentiate individual instances. Recent efforts focus on universal embeddings (Ypsilantis et al., 2023) that cover jointly a whole range of domains and tasks. However, models still require fine-tuning with class-supervised learning to acquire the necessary discriminative properties, making the scarcity of high-quality labeled datasets a major challenge. Data augmentation techniques (Ypsilantis et al., 2021) help mitigate this issue by generating diverse variations of an instance from limited samples. A method leveraging generative models for instance-level tasks (Sundaram et al., 2025) fine-tunes a model per instance, requiring a few real images as input. In contrast, we train a single model that generalizes across objects and domains without relying on real images. Moreover, Kalantidis et al. (2024) uses generative models to augment real images by synthesizing variations that typically distract outdoor visual localization, which is an instance-level task.

**Training with synthetic images** Synthetic data has been used in a variety of computer vision problems, such as object detection (Peng et al., 2015; Rozantsev et al., 2015; Georgakis et al., 2017), segmentation (Chen et al., 2019; Ros et al., 2016), autonomous driving (Abu Alhaija et al., 2018), object pose estimation (Cai et al., 2022; Labbé et al., 2020), 3D-tasks (Chang et al., 2015), and recently for representation learning (Tian et al., 2024; Wu et al., 2023). An early practice is to cut the real objects and paste them onto backgrounds to generate synthetic images for instance or object detection (Dwibedi et al., 2017; Georgakis et al., 2017). However, challenges remain in reducing the boundary artifacts and achieving consistent lighting conditions between the object and background, as these problems often result in unrealistic composite images. More recently, the main sources of synthetic images are computer graphics pipelines or rendering engines (Mahmood et al., 2019), generative adversarial networks (GAN) (Besnier et al., 2020; Brock, 2018), and text-to-image GDM (Fan et al., 2024; Sarıyıldız et al., 2023). Images generated through rendering engines often suffer from domain gap when compared to real-world test images, requiring domain adaptation techniques to mitigate the gap during training. In contrast, GAN and GDM produce more realistic images that do not typically require post-generation domain adaptation (Wang et al., 2020). Text-to-image GDM, in particular, offers a higher degree of control in the image generation process, for example, changing the background of the target object using text prompts (Mokady et al., 2023; Raj et al., 2023; Geng et al., 2024; Zhang et al., 2023). This ability to control image features through text makes GDM particularly valuable for generating diverse images, which is crucial for representation learning (Tian et al., 2024; Wu et al., 2023). However, synthesizing images for instance-level task is not trivial, as it requires generating a synthetic object under various conditions while preserving its structure and texture.

**Metric learning for image retrieval** Given a training dataset, the most common approach for training deep representation networks for image retrieval is supervised learning using categorical labels. As a result, a large number of methods have proposed classification-based losses (Zhai & Wu, 2018; Deng et al., 2019; Teh et al., 2020; Qian et al., 2019; Kim et al., 2020). Despite not directly optimizing the pairwise distance metric that is used at test time, such approaches achieve very good performance, especially when combined with propagating the representation across examples (Elezi et al., 2020; Seidenschwarz et al., 2021; Kotovenko et al., 2023). Other methods directly optimize the distance metric with pairwise losses. These most often rely on hand-crafted loss functions, such as the most popular contrastive (Hadsell et al., 2006), and triplet loss (Schroff et al., 2015), by postulating a correlation between such a training objective and the test time objective which is typically an information retrieval metric. Finding informative pairs and triplets (Musgrave et al., 2020; Roth et al., 2020; Oh Song et al., 2016; Sohn, 2016) appears to be very important. As a natural follow-up, a few recent methods directly optimized differentiable approximations of retrieval metrics, such as average precision (Rolínek et al., 2020; He et al., 2018; Revaud et al., 2019; Ramzi et al., 2021; 2022) and recall (Patel et al., 2022). In this work, we rely on recall@k (Patel et al., 2022) as a loss function which is demonstrating top results on a variety of benchmarks in the literature and does not require hard negative mining. Self-supervised (Kim et al., 2022) methods exist as well and are shown effective, but are tested only on training data from the target distributions, which is not a realistic setup. A recent alternative to CLIP (Radford et al., 2021), called Unicom (An et al., 2023), trains on LAION 400M (Schuhmann et al., 2021), treats captions as weak annotations to perform text-based clustering, and reformulates the learning as a classification task. Their results show improvements in a set of different retrieval datasets, including instance-level ones. Alternatively, we propose leveraging synthetic data to introduce an extensive collection of objects with diverse variations into the training dataset.

## 3 Method

Here, we formulate the target task and describe the training data generation and representation learning. An overview of the proposed generation process is shown in Figure 2.

### 3.1 Task formulation

The target task is instance-level image retrieval. Given a query image, the goal is to retrieve all positive images from a database (db), i.e. those that depict the same object instance as the query. Images depicting different object instances, even if they belong to the same semantic category, are negatives and should not

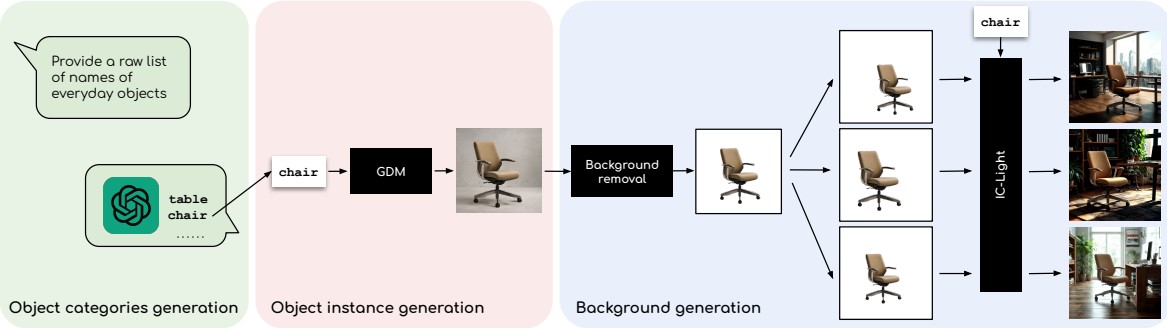

Figure 2: Overview of instance-level training data generation. A domain name or description is the only input, which is used to prompt an LLM to provide a list of object category names. Then, we generate examples of those categories using a GDM, remove the background, and synthesize lighting and background multiple times per generated example to create a diverse set of positive images for each instance.

be retrieved. This is an open-world task, testing on unseen objects from a variety of domains which may be seen or unseen during training.

We consider the efficient retrieval variant using global descriptors. Formally, an image $x$ is mapped to a $d$-dimensional global descriptor $\mathbf{z} = f_\theta(x) \in \mathbb{R}^d$. Retrieval is performed via nearest neighbor search in Euclidean space, ranking database descriptors based on their cosine similarity to the query. The encoder, parameterized by $\theta$, is optimized during training. We focus on fine-tuning foundational models (Zhai et al., 2023) that already perform well by pretraining.

## 3.2 Instance-level training data generation

We propose a pipeline that requires only the name, or a textual description, of a target domain as input, and automatically generates an image training set with instance-level labels. The process consists of four stages: (i) *Objects categories generation* by prompting an LLM to provide a list of object category names; (ii) *Object instance generation* by prompting a GDM to generate object instances from each category; (iii) *Background generation* by synthesizing diverse backgrounds per instance; (iv) *Viewpoint variations* by augmenting the generated images with geometric transformations. Each stage of the process is detailed below.

**Object categories generation**   Object categories (e. g. *table*, *chair*, *clock*) are needed to prompt the GDM for image generation. We automatically obtain a list of object categories by prompting an LLM with minimal information about the domain of interest. In the general case in which we do not target a specific domain, the prompt we use is "Provide a raw list of names of everyday objects." For specific domains, such as artwork, landmark, or product, we enrich the prompt with relevant information and hint with a few examples of object categories. Full details of the designed prompts are provided in the supplementary material. This approach yields a rich and diverse list of $C$ object categories. Examples of category names generated for the general case are *sofa*, *desk*, while for the specific domains are *bust*, *castle*, and *polaroid film*, for artwork, landmark, and product, respectively.

**Object instance generation**   We prompt a GDM, in particular Stable Diffusion Turbo (Sauer et al., 2024), with an object category to generate $K$ images per category. We assume that generating images with different random seeds produces variations that are distinct and recognizable as separate instances within the same category. Therefore, following an instance-level class definition, each of the $M$ generated images, where $M = CK$, is treated as a separate class in our training set. To facilitate the follow-up step of background generation, we target a simple or uniform background. To achieve this, we add "in a clean background" to the prompt after the object category as in, "*a table in a clean background.*" Examples in Figure 3 show that, even though the background removal process may fail in both cases, it is less likely to happen with the extended prompt, while the original prompt provides outputs with richer background.

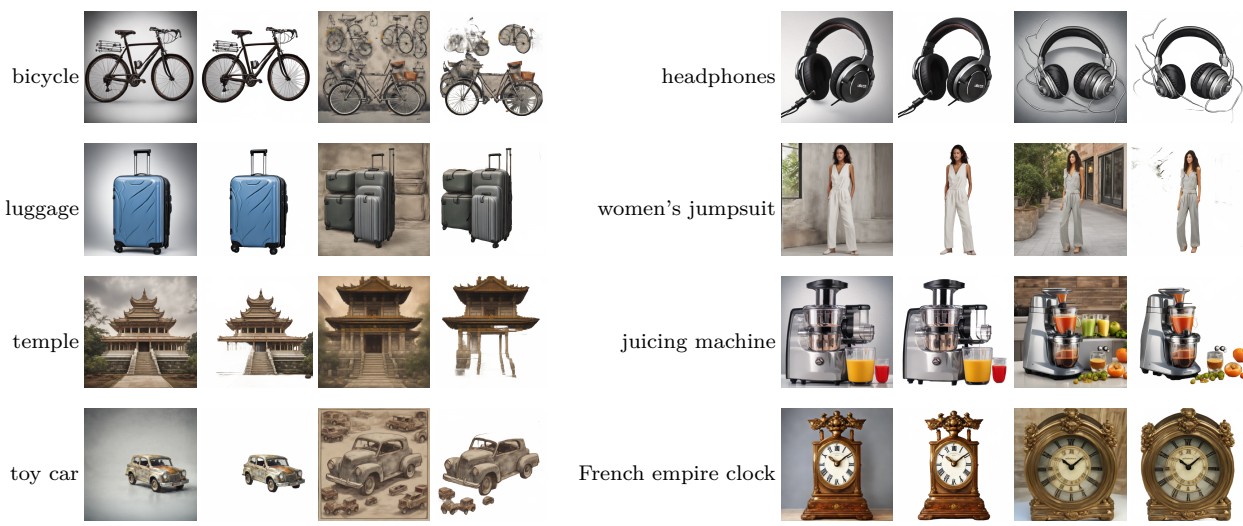

Figure 3: Examples of object instances generated by GDM for specific categories. We show the category name, the generated image and the background removal process using "in a clean background" (columns 1 & 2) and without it (columns 3 & 4).

**Background generation** We create variations of an object instance by generating images with multiple, distinct backgrounds and lighting conditions. Given a generated instance in the previous step, we rely on ICLight (Zhang et al., 2025) to perform the relighting and add different backgrounds. This process is conducted in three parts:

1. **Background removal**. Firstly, the background of the generated instance is removed to ensure that the input image only depicts the object of interest. We use RMBG v1.4[1], which relies on the IS-Net model (Qin et al., 2022), which is a model trained on a large dataset of background-foreground masks. The RMBG tool outputs a soft alpha matte indicating foreground probability. The background is removed by alpha-blending the image with a constant background color, resulting in smooth object boundaries.

2. **Size and position variation**. We additionally perform padding of random length/width and resize to the original resolution so that the object appears at different sizes and positions. Each left/right (top/bottom) pad is up to half of the image width/height. To preserve the aspect ratio and avoid distortion, we first sample the total horizontal padding, i. e. , the sum of the left and right padding, compute the corresponding total vertical padding using a scaling factor, and adjust the top and bottom padding accordingly. The padding area is filled with the color of pixel (0, 0) from the image with the background removed.

3. **New background and lightning**. Then, the object category is used as a prompt to guide ICLight to generate an environment that is commonly appropriate for the specific object. ICLight is designed to produce realistic lighting on the object and maintain consistent illumination between the object and the background. The semantic coherence between object and background, as well as the faithfulness of the background to the prompt, is largely inherited from the underlying pre-trained Stable Diffusion v1-5[2]. We apply ICLight with its default settings.

We repeat the last two parts $N$ times per generated object instance with different seeds to generate multiple backgrounds. The $N$ images are all elements of the same class in our training set and the only members of this class. Figure 4 shows examples of generated lighting and background for a variety of object categories.

---

[1] https://huggingface.co/briaai/RMBG-1.4
[2] https://huggingface.co/stablediffusionapi/realistic-vision-v51

Table 1: Statistics of the generated training dataset. ILGen-G and ILGen-S comprise only objects from the generic domain and one of the specific domains, respectively. ILGen-ALL comprises 50% of objects from the generic domain (10K) and all objects from the three specific domains (10K), i. e. 20K objects in total.

| domain of objects | $C$ | $K$ | instances |
|---|---|---|---|
| generic | 2,000 | 10 | 20,000 |
| art | 200 | 15 | 3,000 |
| landmark | 50 | 80 | 4,000 |
| product | 200 | 15 | 3,000 |

**Viewpoint variations** All images of a class depict the object under different backgrounds and similar viewpoints which only varies because of the padding of the previous step. We additionally rely on simple random geometric augmentations during training to further modify the object's geometry. This process resembles self-supervised learning with instance-discrimination (Oquab et al., 2023; Chen et al., 2020), where two positive examples are just two different random augmentations of the same input image. Nevertheless, there is an essential difference in our case, that the background and lighting significantly vary. Such a factor makes our training setting a unique of its kind. We additionally explore synthesizing viewpoints by rendering multiple angles of GDM-generated instances, then applying background generation for each rendered view. Details are presented in the supplementary.

### 3.3 Representation learning

In total, our generated dataset contains $CKN$ training images, forming $CK$ classes coming from $C$ object categories. We construct training batches by sampling $B$ classes and all their corresponding images, resulting in $NB$ images per batch. During training, we adopt a query v. s. database scheme: one image from each of the $N$ images per class is randomly chosen as the query, while the remaining $NB - 1$ images of the batch form the database, as shown in Figure 5.

The similarity between the query and db images is computed in $\hat{\mathbf{y}} \in \mathbb{R}^{NB-1}$, while $\mathbf{y} \in \{0,1\}^{NB-1}$ denotes the labels of all db images with respect to the query, i. e. positive or negatives based on their classes. We optimize an information retrieval metric as the loss function, in particular an approximation of recall at the top-$k$ ranks, based on $\hat{\mathbf{y}}$, and $\mathbf{y}$. We train with the average of recall@k loss estimated for different values of $k$. The approximation of recall is possible by formulating its estimation with the use of step functions, which, during training, are replaced with a sigmoid function. The technical and implementation details can be found in the original paper (Patel et al., 2022).

## 4 Experiments

### 4.1 Experimental details

**Data generation details** We use GPT-4o (Hurst et al., 2024) as an LLM for the object categories generation. The LLM is prompted to generate two types of objects: generic and domain-specific. Generic objects consist of daily-life objects, while domain-specific objects are objects represented in the particular domains of our evaluation benchmarks. Details about the number of generated object categories are in Table 1. Since the generic domain broadly covers concepts spanning across several specific domains, we assign it a larger number of object categories ($C$) and use smaller numbers for specific domains given their narrower scope. To balance the total number of instances, we assign different numbers of instances per category ($K$) accordingly. See supplementary for ablation study on $C$ and $K$. We set the number of inference steps to 1 when generating instances from each object category using Stable Diffusion Turbo. Before applying ICLight to synthesize four distinct backgrounds, i. e. $N = 4$, we add random padding (up to 50% of the image resolution) to the foreground-segmented instance, keeping the same aspect ratio.

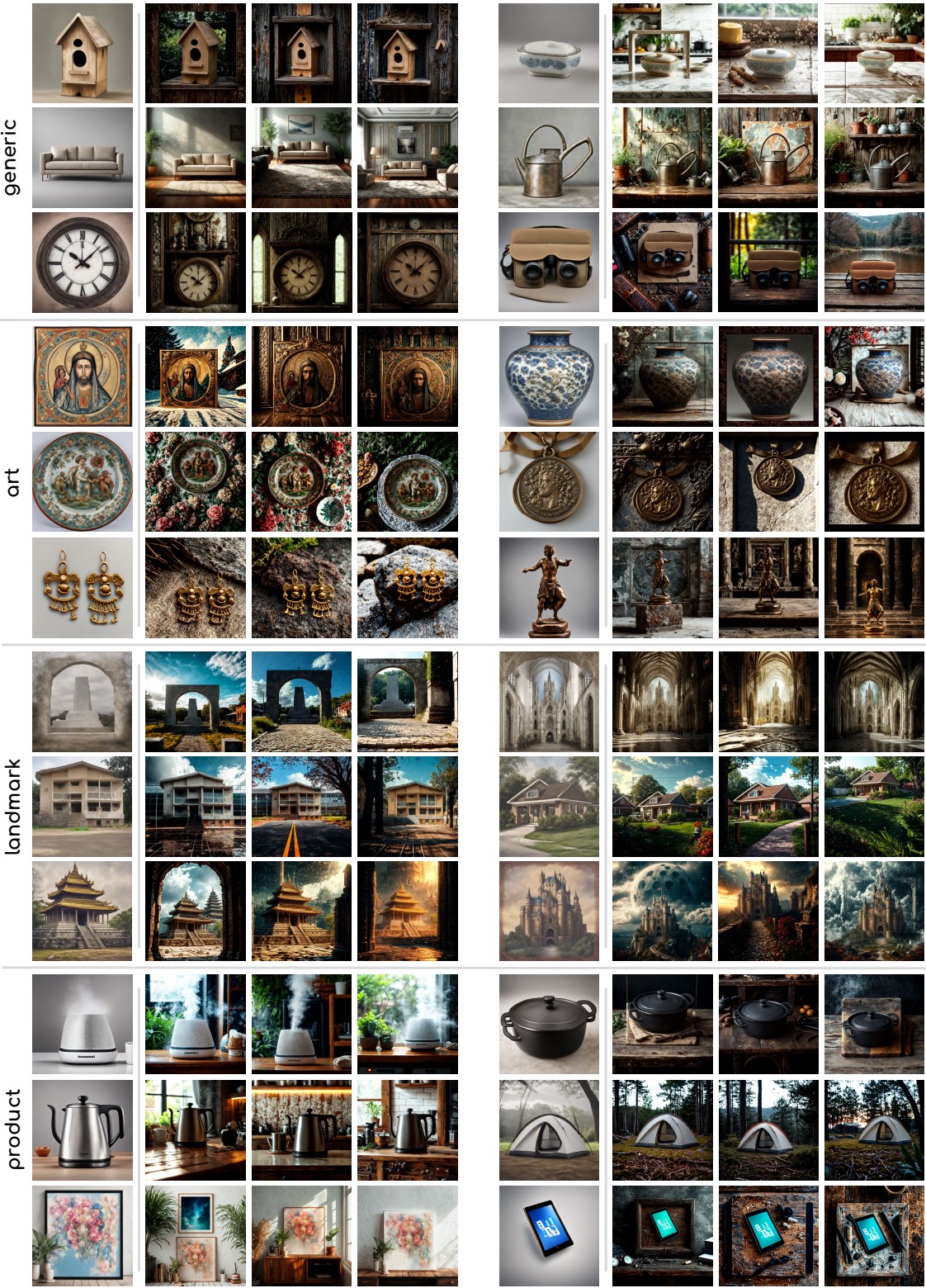

Figure 4: Examples of object instances generated by GDM (column 1), and the generated images that leave the object intact and add lighting and background that is well suited to the object (columns 2 ∼ 4).

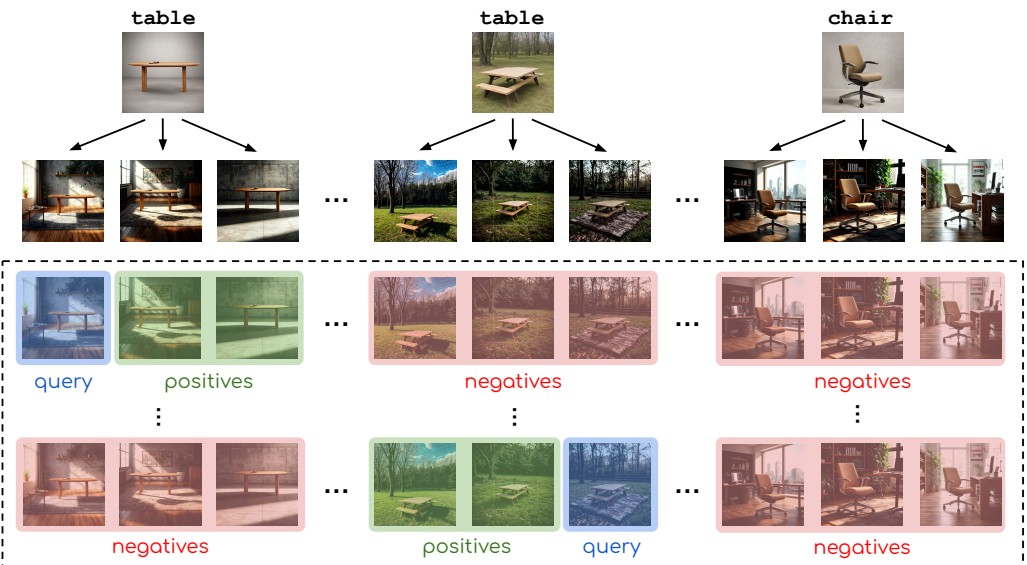

Figure 5: Training batch construction for instance-level representation learning. A batch simulates a retrieval task with a query (blue) and database of positive (green) and negative (red) images. Images are considered positive if they belong to the same class, otherwise they are negatives. An image encoder is trained with metric learning on this batch.

Table 2: Details of evaluation datasets.

| dataset | queries | database | domain | metric |
|---|---|---|---|---|
| MET (Ypsilantis et al., 2021) | 19.3K | 224.4K | artwork | mAP@100 |
| R-Oxford (Radenović et al., 2018) | 70 | 5.0K + 1M | landmark | mAP |
| R-Paris (Radenović et al., 2018) | 70 | 6.3K + 1M | landmark | mAP |
| GLDv2 (Weyand et al., 2020) | 1.1K | 761.8K | landmark | mAP@100 |
| SOP (Oh Song et al., 2016) | 60.5K | 60.5K | product | mAP@100 |
| INSTRE (Wang & Jiang, 2015) | 1.3K | 27.3K | multi | mAP |
| mini-ILIAS (Kordopatis-Zilos et al., 2025) | 1.2K | 4.7K + 5M | multi | mAP@1K |

**Training set variants** To evaluate the quality of our generated data, we compare the performance of the backbone models trained on our generated dataset, some of its variants and alternatives with real objects and/or images.

- **Pretrained**: The original datasets which the backbones are pretrained on. SigLIP and CLIP are pretrained on web-based text-image datasets, WebLI (Chen et al., 2023) and WIT (Radford et al., 2021), respectively. ViT is pretrained on ImageNet (Dong et al., 2009). The frozen backbones are evaluated.

- **ILGen-ALL - all domains**: Our generated dataset with 10K objects from the generic domain and 10K objects from the specific domains. This dataset is used by default, unless otherwise stated. See Table 1 for details.

- **ILGen-G - generic domain**: Our generated dataset with up to 20K objects from the generic domain only.

- **ILGen-S - specific domain**: Our generated dataset with images from only one of the three specific domains.

- **ILGen-ALL without background**: Our generated dataset without background generation.

Table 3: Evaluation results using SigLIP with different training datasets, number of instances, and use of synthetic background (bg). ILGen-G uses generic domain object categories, while ILGen-ALL includes domain-specific objects. We train each setting for 3 seeds, and report the mean and standard deviation.

| ID | data | instance | avg | artwork | landmark | | product | multi | |
|---|---|---|---|---|---|---|---|---|---|
| | | | | MET | ROP | GLD | SOP | INS | mIL |
| 1 | pretrained | - | 47.5 | 67.3 | 45.0 | 15.7 | 55.4 | 80.6 | 21.0 |
| 2 | Objaverse-background | 20K | 50.9±0.5 | 74.1±0.1 | 42.6±1.0 | 15.9±0.5 | 57.6±0.4 | 86.8±0.5 | 28.6±1.1 |
| 3 | ILGen-G | 5K | 51.2±0.3 | 72.8±0.6 | 46.5±0.3 | 17.4±0.1 | 55.3±0.3 | 85.9±0.3 | 29.4±1.6 |
| 4 | ILGen-G | 10K | 51.6±0.2 | 72.7±0.1 | 46.3±0.2 | 17.7±0.2 | 55.4±0.1 | 87.1±0.0 | 30.2±1.0 |
| 5 | ILGen-G | 20K | 50.8±0.2 | 72.4±0.2 | 46.4±0.4 | 17.4±0.3 | 55.9±0.1 | 85.4±0.1 | 27.4±1.1 |
| 6 | ILGen-ALL w/o bg | 20K | 49.5±0.3 | 73.0±0.6 | 46.8±1.2 | 17.4±0.2 | **61.1**±0.5 | 77.3±0.6 | 21.5±1.0 |
| 7 | ILGen-ALL | 20K | **52.7**±0.3 | **75.1**±0.2 | **48.6**±0.3 | **18.7**±0.4 | 55.6±0.3 | **87.5**±0.4 | **30.6**±0.6 |

- **Objaverse-background**: Objaverse 1.0 (Deitke et al., 2023) is a large-scale 3D object dataset with 818K 3D objects across various categories. We randomly select 20K objects, render each 3D object into 16 views (Liu et al., 2024), and choose the four views around the main one, resulting in a total of 80K images to match the statistics of our generated dataset. For each view, we add a background with the same generation process as in our method. This dataset allows us to compare with training on real objects rather than synthetic ones, but on synthesized images via rendering.

- **Real-S - specific domain**: To compare with training on real images that are manually annotated, we use the MET, GLDv2, and SOP training sets to obtain domain-specific models for artwork, landmark, and product, respectively. We follow the same dataset split as in Ypsilantis et al. (2023). To provide a direct comparison, we use the same number of instances as the corresponding domain-specific parts of our dataset, i.e. 3K, 4K, and 3K, respectively, and 4 images per instance.

- **Real-ALL - all domains**: The above is extended to compose a dataset by merging the training sets of SOP, InShop, RP2k, GLDv2, and MET. We use all classes with at least 4 images from the first three datasets that are small, and complement with enough classes equally from the other two datasets to reach 20K instances. We sample 4 images per class.

**Training details**   During training, we use random cropping, resizing, flipping, color jitter, and mapping to grayscale as image augmentations (He et al., 2020). We use a batch size of $1,600$ images ($B = 400$, $N = 4$) and optimize over 400 queries, one per class. We use the vanilla version of the recall@k loss with its default hyper-parameters, $k = \{1, 2, 4, 8\}$ for the recall@k loss with the two temperatures set to 0.01 and 1.0 as in the original work, and train until all classes have been loaded in a batch. We use learning rate $10^{-5}$ and Adam optimizer (Kingma & Ba, 2015) with a weight decay $10^{-6}$. Experiments are run on a single A100 or V100 GPU. The training process of ILGen-ALL with SigLIP takes approximately 2.5 hours on an A100 GPU.

**Backbones**   We use SigLIP ViT-L/16 (Zhai et al., 2023), CLIP ViT-L/14 (Radford et al., 2021), and ViT-B/16 (Dosovitskiy et al., 2021), briefly referred to as SigLIP, CLIP, and ViT-B. Images are resized to $336 \times 336$, $384 \times 384$, and $224 \times 224$ pixels, respectively, according to their pretraining setup. We load the pre-trained models from timm[3] and treat the [CLS] token as the global descriptor.

**Evaluation benchmarks**   We use a set of standard and diverse ILR retrieval and classification datasets for evaluation. ILR datasets are comprised of queries, a database in which the same instances as queries exist as positives, and occasionally, a distractor set of irrelevant images. Details are provided in Table 2 and the dataset list is as follows:

---

[3]https://timm.fast.ai/

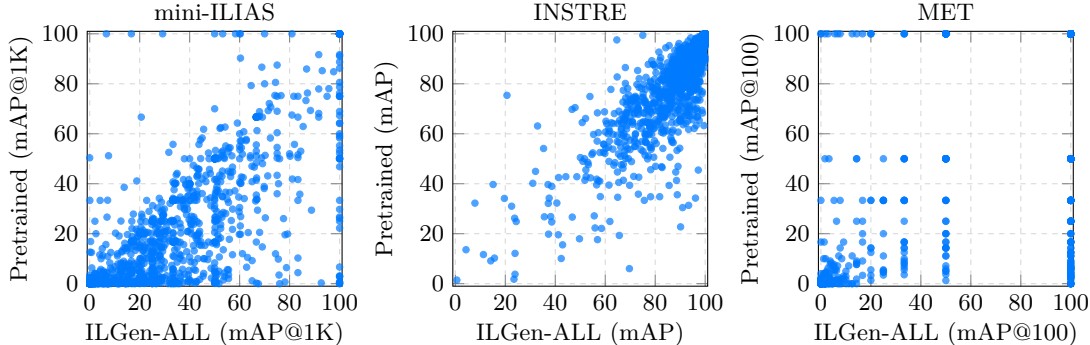

Figure 6: Average Precision (AP) per query for the pretrained backbone (y-axis) and the backbone fine-tuned on ILGen-ALL (ID7) (x-axis). Each point represents a query in the evaluation dataset. Points below the diagonal indicate a query with improved performance when fine-tuned on ILGen-ALL. Results using SigLIP.

- **Artwork domain**: The MET dataset (Ypsilantis et al., 2021) comprises a database of catalog photos from the Metropolitan Museum of Art and query images taken by visitors inside the museum. To adapt the benchmark for retrieval, we retain only queries with at least one positive match in the database, i. e. we discard the distractor queries, and keep only the first positive per query in the database asserting visual overlap between the two images.

- **Landmark domain**: R-Oxford (Radenović et al., 2018), R-Paris (Radenović et al., 2018), and GLDv2 (Weyand et al., 2020) are the most widely used datasets in this domain. For R-Oxford and R-Paris, we report results on the Medium and Hard evaluation split with 1M distractors, and following standard practice, we report average performance across the two datasets, denoted as ROP.

- **Product domain**: SOP (Oh Song et al., 2016) whose images are crawled from e-commerce websites.

- **Multi-domain**: We use INSTRE (Wang & Jiang, 2015) and ILIAS (Kordopatis-Zilos et al., 2025) which include a variety of objects from multiple domains such as daily objects, landmarks, etc. We use the mini version of ILIAS with 5M distractor images.

## 4.2 Results for different training sets

Table 3 shows the main results for SigLIP after training on a variety of datasets.

**Impact of synthetic data**  ILGen-ALL (ID7) provides consistent improvement compared to the pretrained (ID1) model on all datasets, with an average improvement equal to 5.2. Compared to Objaverse, which uses images rendered from 3D objects rather than automatically generated, ILGen-ALL performs better on most datasets, especially on SOP. This suggests that our method, which relies solely on synthesized objects, learns representations that are at least as effective as those learned on rendered objects.

**Number of instances**  We evaluate SigLIP backbone trained on the generic-domain version of the dataset, ILGen-G, with different numbers of instances: 5K, 10K, and 20K (corresponding to ID3, ID4, and ID5 in Table 3). Even with the smallest set of 5K generic instances (ID3), performance on all the benchmarks is better than the pre-trained backbone (ID1) except SOP where performance does not change. When the number of instances increases to 10K (ID4), the average performance increases further, but saturates for the largest set (ID5).

**Diverse v. s. clean background**  Training on ILGen-ALL with clean background (ID6) improves the performance on most datasets compared to the pretrained backbone. However, performance drops on IN-STRE and the improvement is small on mini-ILIAS, which are two datasets with high background clutter. Synthesizing realistic and diverse backgrounds (ID7) leads to a substantial improvement on most datasets compared to clean background (ID6). SOP forms an exception, where having clean background is the variant that brings a noticeable improvement, which is related to the commonly clean background in this test set.

Table 4: Comparison between training on real-labeled images and training our synthetic images on four different domains using SigLIP.

| ID | dataset | avg | artwork | landmark | | product | multi | |
|----|---------|-----|---------|----------|--|---------|-------|--|
| | | | MET | ROP | GLD | SOP | INS | mIL |
| 1 | pretrained | 47.5 | 67.3 | 45.0 | 15.7 | 55.4 | 80.6 | 21.0 |
| 8 | Real-S (artwork) | 49.9 | **75.2** | 46.8 | 17.18 | **57.0** | 80.9 | 22.4 |
| 9 | ILGen-S (artwork) | **51.2** | 73.7 | **47.0** | **17.24** | 55.6 | **85.4** | **28.3** |
| 10 | Real-S (landmark) | 50.0 | 69.6 | **55.0** | **19.8** | 56.7 | 78.6 | 20.2 |
| 11 | ILGen-S (landmark) | **51.0** | **72.5** | 50.7 | 19.7 | 54.6 | **84.4** | **24.2** |
| 12 | Real-S (product) | 48.3 | 63.8 | 46.1 | 16.9 | **60.3** | 80.7 | 21.8 |
| 13 | ILGen-S (product) | **50.5** | **71.6** | **46.4** | **17.0** | 55.9 | **85.0** | **27.1** |
| 14 | Real-ALL | 51.4 | 69.3 | **55.3** | **19.7** | **71.8** | 72.7 | 19.3 |
| 7 | ILGen-ALL | **52.7** | **75.1** | 48.6 | 18.7 | 55.6 | **87.5** | **30.6** |

Table 5: Training on a mixture of real and synthetic data (ID S25-30) performs better than training solely on real (ID14) or synthetic data (ID7).

| ID | training instances | | avg | artwork | landmark | | product | multi | |
|----|--------------------|--|-----|---------|----------|--|---------|-------|--|
| | ILGen-ALL | real | | MET | ROP | GLD | SOP | INS | mIL |
| 1 | - | - | 47.5 | 67.3 | 45.0 | 15.7 | 55.4 | 80.6 | 21.0 |
| 14 | - | 20,000 | 51.4 | 69.3 | 55.3 | 19.7 | 71.8 | 72.7 | 19.3 |
| 7 | 20,000 | - | 52.7 | 75.1 | 48.6 | 18.7 | 55.6 | 87.5 | 30.6 |
| S25 | 20,000 | 2,000 | 53.5 | 76.3 | 51.0 | 19.4 | 58.3 | 86.3 | 29.7 |
| S26 | 20,000 | 4,000 | 53.9 | 76.8 | 51.1 | 19.4 | 60.8 | 86.2 | 29.4 |
| S27 | 20,000 | 8,000 | 54.2 | 75.6 | 53.7 | 18.7 | 64.7 | 84.5 | 27.7 |
| S28 | 20,000 | 12,000 | 55.3 | 75.2 | 55.0 | 20.2 | 66.7 | 85.5 | 29.2 |
| S29 | 20,000 | 16,000 | 54.0 | 73.3 | 54.1 | 18.8 | 69.2 | 82.7 | 25.9 |
| S30 | 20,000 | 20,000 | 55.0 | 75.3 | 56.5 | 20.4 | 70.9 | 81.4 | 25.4 |

**Domain of the instances** Complementing ILGen-G-10K (ID4) with 10K images from domain-specific objects (ID7) is much better on average than complementing it with 10K generic objects (ID5). Such a choice strengthens performance on all test sets across domains except SOP Therefore, leveraging synthetic images in a diverse set of targeted domains, our method has the potential to effectively address data scarcity and obtain universal representation models.

**Improvement per query** In Figure 6, we compare the performance of the pretrained and the fine-tuned SigLIP on ILGen-ALL (ID7) on a query basis. Training on the dataset of the proposed method improves the performance on the majority of queries and over the whole range of performance values with the pretrained model, even for many highly performing queries of INSTRE.

**Comparison to real manually labeled images** We train SigLIP on both real-labeled and our synthetic images with recall@k loss under the same setting and present results in Table 4. We make the following observations. Training with our synthetic images yields better overall performance compared to real-labeled images. Although training with real images from a single domain achieves better performance within the specific domain, our synthetic images have better performance across other domains except for product. Notably, results on multi-domain (INSTRE and mini-ILIAS) reveal that our synthetic images are the best in all cases, indicating the strength of our approach to cover a large range of domains. Performance when testing on ROP is always better when training on real images, possibly indicating shortcomings of the generative models for large objects with many details.

Table 6: Evaluation results on different backbones. Representations learned on synthetic data using ILGen-ALL outperform the pretrained representations on all datasets, except ViT on SOP.

| model | data | avg | artwork | | landmark | product | multi | |
| | | | MET | ROP | GLD | SOP | INS | mIL |
|---|---|---|---|---|---|---|---|---|
| SigLIP | pretrained | 47.5 | 67.3 | 45.0 | 15.7 | 55.4 | 80.6 | 21.0 |
| | ILGen-ALL | **52.7** | **75.1** | **48.6** | **18.7** | **55.6** | **87.5** | **30.6** |
| CLIP | pretrained | 37.5 | 47.1 | 40.0 | 10.5 | 41.8 | 75.1 | 10.4 |
| | ILGen-ALL | **46.8** | **69.6** | **43.7** | **16.8** | **45.5** | **81.7** | **23.8** |
| ViT-B | pretrained | 25.7 | 34.2 | 24.6 | 5.7 | **43.7** | 41.9 | 4.0 |
| | ILGen-ALL | **34.3** | **50.8** | **29.8** | **9.1** | 40.8 | **65.1** | **10.1** |

Table 7: Evaluation results by training SigLIP on ILGen-ALL using different loss function.

| loss | avg | artwork | | landmark | product | multi | |
| | | MET | ROP | GLD | SOP | INS | mIL |
|---|---|---|---|---|---|---|---|
| pretrained | 47.5 | 67.3 | 45.0 | 15.7 | 55.4 | 80.6 | 21.0 |
| recall@k (Patel et al., 2022) | **52.7** | **75.10** | **48.56** | 18.7 | **55.6** | 87.5 | 30.6 |
| infoNCE (Chen et al., 2020) | 52.2 | 75.06 | 48.55 | **18.8** | 54.2 | 86.0 | 30.7 |
| contrastive (Chopra et al., 2005) | 50.6 | 62.8 | 46.0 | 16.1 | 53.8 | 86.4 | **38.4** |
| softmax margin (Wang et al., 2018) | 51.5 | 70.1 | 47.3 | 18.4 | 55.4 | **88.0** | 29.8 |

**Training with both real and synthetic data** Table 5 presents results for progressively adding real data on top of synthetic data (ID-S25 to ID-S29), up to using all the real data (ID-S30). Overall, in terms of average performance across all datasets, the mixture of real and synthetic training data performs better than using only real or only synthetic data. This holds for all proportions of mixing the data, with better overall performance for balanced (ID-S30) or nearly balanced (ID-S28) mixing. In domains where training data is rich and diverse the mixed data performs slightly worse than using only real images; products being the only such case. This experiment indicates that synthetic data serves, not only as a replacement, but also as an effective complement to real data for instance-level learning across various domains.

## 4.3 Ablations and more results

**Backbones** In Table 6 we present results for fine-tuning two additional backbones. Performance improvements are similar to those of SigLIP, demonstrating the general applicability of our method.

**Different loss function** We train SigLIP using infoNCE loss (Chen et al., 2020), contrastive loss (Chopra et al., 2005), and softmax margin loss (Wang et al., 2018), which are widely used in representation learning. The generated training set is shown to be effective with a diverse set of losses, while the recall@k loss remains the best overall choice. Implementation details for this experiment are as follows. For infoNCE loss, we follow the same strategy for batch construction and training parameters as for the recall@k loss. Each image acts as one query, and its positive and negative pairs depend on the instance-level class label. We use a 0.05 temperature during training. Regarding the contrastive loss, we treat each image as a query and randomly sample one positive among images with the same instance-level class label. For the negatives, we mine the hardest one in the dataset based on the cosine similarity of the image descriptors. We use a margin of 1. The learning rate is $10^{-7}$, and the batch size is 8. For softmax margin loss, we follow the training process proposed in UnED (Ypsilantis et al., 2021), the backbone remains frozen for the first two epochs, and only the classifier is trained with a learning rate of $10^{-3}$. In the following epochs, the network is trained end-to-end with a $10^{-6}$ learning rate. Since this is a classification loss, no specific curation of the batches is necessary. We use a batch size of 16. The scale and margin parameters are set at 16 and 0, respectively, as in UnED. The results are present in Table 7.

Table 8: Ablation study on training data (S1-S2), LLM (S3-S5), GDM (S6-S7), and background generation (S8-S9). ID1 (pretrained) and ID7 (ILGen-ALL) were presented in Table 3. Each ablation modifies only one component compared to ID7. *Pos* refers to the number of training images per instance class. *Steps* are the inference steps during image generation. SD Turbo uses 1 step by default. SD refers to Stable Diffusion.

| ID | data | | LLM | | GDM | | background | | results | | | | | | |
|----|---------|-----|----------|----------|---------|-------|---------|---------|------|------|------|------|------|------|------|
| | dataset | pos | prompt | model | model | steps | model | padding | avg | MET | ROP | GLD | SOP | INS | mIL |
| 1 | pretrained | - | - | - | - | - | - | - | 47.5 | 67.3 | 45.0 | 15.7 | 55.4 | 80.6 | 21.0 |
| S1 | generated | 3 | designed | GPT-4o | SD Turbo | 1 | ICLight | ✔ | 51.5 | 73.7 | 47.8 | 18.3 | 56.2 | 85.6 | 27.7 |
| S2 | generated | 2 | designed | GPT-4o | SD Turbo | 1 | ICLight | ✔ | 50.3 | 71.3 | 46.6 | 17.8 | 55.3 | 85.5 | 25.2 |
| S3 | generated | 4 | template | GPT-4o | SD Turbo | 1 | ICLight | ✔ | 52.6 | 74.9 | 48.0 | 18.6 | 56.5 | 86.2 | 31.6 |
| S4 | generated | 4 | designed | DeepSeek | SD Turbo | 1 | ICLight | ✔ | 52.6 | 75.3 | 47.0 | 18.2 | 55.0 | 88.0 | 32.1 |
| S5 | generated | 4 | designed | Claude | SD Turbo | 1 | ICLight | ✔ | 52.5 | 74.6 | 48.8 | 18.3 | 55.5 | 87.5 | 30.6 |
| S6 | generated | 4 | designed | GPT-4o | SD v2.0 | 50 | ICLight | ✔ | 51.8 | 74.1 | 47.6 | 18.2 | 56.6 | 86.9 | 27.6 |
| S7 | generated | 4 | designed | GPT-4o | SD Turbo | 5 | ICLight | ✔ | 53.0 | 74.7 | 49.1 | 18.2 | 56.6 | 88.3 | 31.0 |
| S8 | generated | 4 | designed | GPT-4o | SD Turbo | 1 | SD v2.0 | ✔ | 47.1 | 70.8 | 48.8 | 17.8 | 54.6 | 75.4 | 15.2 |
| S9 | generated | 4 | designed | GPT-4o | SD Turbo | 1 | ICLight | ✗ | 51.5 | 75.1 | 49.9 | 19.1 | 57.4 | 82.9 | 24.7 |
| 7 | generated | 4 | designed | GPT-4o | SD Turbo | 1 | ICLight | ✔ | 52.7 | 75.1 | 48.6 | 18.7 | 55.6 | 87.5 | 30.6 |

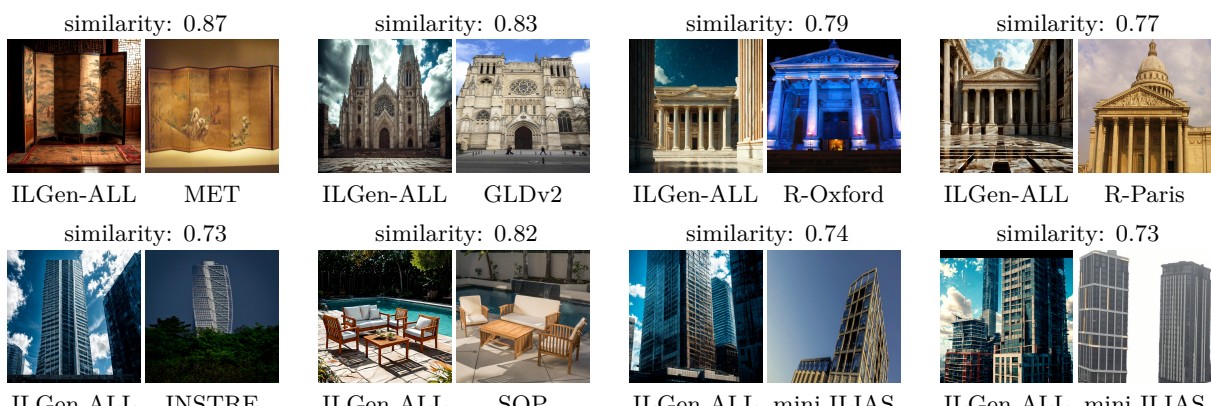

similarity: 0.87 — ILGen-ALL / MET  •  similarity: 0.83 — ILGen-ALL / GLDv2  •  similarity: 0.79 — ILGen-ALL / R-Oxford  •  similarity: 0.77 — ILGen-ALL / R-Paris

similarity: 0.73 — ILGen-ALL / INSTRE  •  similarity: 0.82 — ILGen-ALL / SOP  •  similarity: 0.74 — ILGen-ALL / mini-ILIAS  •  similarity: 0.73 — ILGen-ALL / mini-ILIAS

Figure 7: Pairs of ILGen-ALL and test sets with the highest similarity score. While these pairs share some common appearance, they do not indicate data leakage from an ILR point of view.

**Training images per class**  Table 8 shows the performance with different numbers of images per instance-level class during training (ID-S1 and ID-S2). We decrease the number of images per class $N$ in the training set to 3 and 2. The trained models achieve an average performance of 51.5 and 50.3, respectively, which is a considerable drop compared to the main variant that achieves 52.7.

**LLM models and prompts**  To examine the effect of the prompts and LLMs, we evaluate variants from ID-S3 to ID-S5 in Table 8. In ID-S3, we use a fixed prompt template across all the generic and specific domains with GPT-4o (see the supplementary material). In ID-S4 and ID-S5, we use our designed prompts with two other LLMs, DeepSeek-V3 and Claude 3.7 Sonnet, respectively. The similar results suggest that our method is robust regardless of the LLM or prompt type.

**GDM**  We apply different GDMs and higher-quality images to study how instance generation quality affects performance, as shown in ID-S6 and ID-S7 in Table 8. In ID-S6, we change SD Turbo to SD v2.0, resulting in worse performance, likely due to more intricate backgrounds that hinder accurate foreground segmentation. We use 50 inference steps following the default setting. In ID-S7, we increase the inference steps of SD Turbo from the default 1 to 5, aiming to generate higher-quality images. Although the visual quality is better, there was no overall significant performance improvement. Additional details are in the supplementary material.

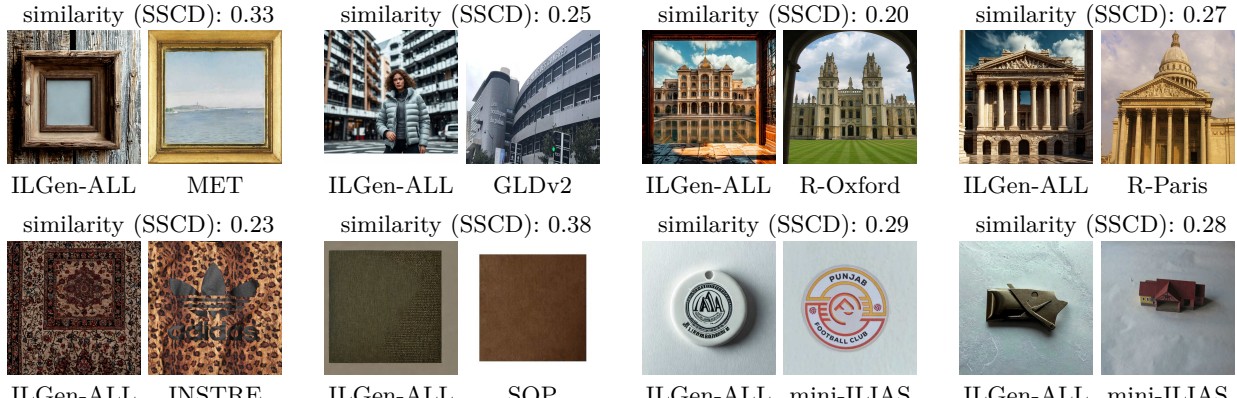

Figure 8: Pairs of ILGen-ALL and test sets with the highest similarity score computed using SSCD. The low similarities indicate no copying between the generated and real images.

**Background generation**  As shown in Table 8, changing ICLight to SD v2.0 for background generation (ID-S8) leads to worse performance even than the pretrained model (ID1). This is due to poor identity preservation, while ICLight is tailored to this task. When we switch off padding (ID-S9), which is our way of varying object size and position, the average performance drops by 1.2, demonstrating that even such a simple viewpoint variation has a positive impact.

**Train and test set overlap**  To investigate whether objects from the test sets have leaked into the generated training set, we perform the following mining process. We first use the trained model (ID7) as a descriptor extractor and perform retrieval using the test queries as queries and the generated training set as the database. We visually inspect the results with the highest similarity scores and do not identify any cases of such leakage as shown in Figure 7. The pairs showcase similar characteristics (a strength of our approach), but are not positive from an instance-level point of view. Moreover, we further use SSCD[4] (Pizzi et al., 2022), specifically the `sscd_disc_large` (ResNeXt101) model for copy detection. Figure 8 shows the pairs with highest similarities between ILGen-ALL and test datasets. The highest similarity score is 0.38, which is strictly below the copy-detection threshold of 0.5 following Somepalli et al. (2023), indicating that none of the generated images are copies of the test images.

# 5 Conclusion

This work introduces a novel approach to training ILR models using generative diffusion models to automatically create diverse, instance-specific training images. By eliminating the need for extensive data collection and curation, our method opens up new opportunities to easily train ILR models across various domains. Although foundational representation models are generally considered universal and capable of performing well across a wide range of domains, we show that fine-tuning these models exclusively on synthetic instance-level data results in notable performance improvements.

### Acknowledgments

This work was partly supported by Junior Star GACR GM 21-28830M, Horizon MSCA-PF grant No. 101154126, and JSPS KAKENHI No. 22K12091 and No. 23H00497. We are grateful to Patrick Ramos, Ryan Ramos, and Lu Wei for their time and effort in the human verification. We also thank Pavel Suma for the support on the code.

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
