# OpenReview forum: "Instance-Level Generation for Representation Learning"
_TMLR — Decision pending for TMLR_

### Review · Reviewer_Vjgr · 2026-01-05

**Summary Of Contributions:**

## Summary
The paper proposes a novel data generation pipeline tailored for Instance-Level Recognition (ILR) that creates a large-scale training set without relying on any real images. The method utilizes a Large Language Model (LLM) to generate lists of object categories for specific domains (e.g., artwork, landmarks, products) and a Generative Diffusion Model (GDM) to synthesize distinct object instances based on different random seeds. To ensure diversity, the pipeline incorporates background removal and lighting/background variations using ICLight. The authors demonstrate that fine-tuning foundation models (e.g., SigLIP) on this purely synthetic dataset leads to performance improvements across several ILR benchmarks compared to pretrained baselines.

## Strengths
- **Clarity and Simplicity:** The paper is written in a clear and accessible manner, making the proposed methodology easy to follow. The pipeline offers a straightforward solution to the complex problem of collecting instance-level data.
- **Effectiveness of Synthetic Data:** The authors successfully demonstrate that training solely with synthetic data generated through their pipeline yields performance improvements over pretrained models, validating the potential of generative models in addressing data scarcity for ILR.

## Weaknesses
- **Ambiguity in Instance Definition:** The core assumption that generating images with different random seeds consistently produces distinct instances (class separation) lacks sufficient analysis. Relying on random seeds to define instance identity appears to be a heuristic tailored specifically for this synthetic generation pipeline, rather than ensuring the semantic or physical distinctiveness required for robust ILR tasks. Consequently, the paper would benefit from a deeper empirical or visual investigation into whether these seed-based variations truly represent distinct instances at a granular level.

- **Reproducibility Concerns:** The paper does not provide the complete list of object names generated by the LLM for the generic, art, landmark, and product domains. Moreover, specific technical details regarding the 'background removal' process and the 'padding' method are missing. While the paper mentions that background removal is conducted and random padding is applied, it fails to specify the exact tools (e.g., specific segmentation models) or algorithms used. Without these implementation details and the specific category lists, reproducing the exact dataset and experimental results is challenging.
- **Lack of Hyperparameter Justification:** In Table 1, the values for the number of object categories ($C$) and instances per category ($K$) vary significantly across domains (e.g., Generic: $C=2000, K=10$ and Product: $C=200, K=15$). The paper lacks a detailed explanation or analysis justifying these specific hyperparameter choices.
- **Limited Novelty and Impact:** The proposed method appears somewhat trivial as it combines existing generative tools without significant technical innovation. More critically, Table 4 indicates that training on domain-specific real data (Real-S) outperforms the proposed synthetic method (ILGen-S), particularly in the landmark and artwork domains. This suggests that synthetic data alone is currently insufficient to fully address the challenges of ILR. The paper's contribution would be significantly stronger if it demonstrated that combining the proposed synthetic dataset with real-world data leads to State-of-the-Art (SoTA) performance, rather than positioning the method solely as an easier data acquisition alternative that yields lower performance than real data. Additionally, a comparison with other synthetic data generation methods mentioned in the Related Work section is missing.

**Audience:**

Yes

**Audience Explanation:**

Given the interest in research utilizing synthetic data, this paper addresses a timely topic and is likely to garner significant attention from the readership.

**Claims And Evidence:**

No

**Claims Explanation:**

Please see the weaknesses.

**Requested Changes:**

Please address my concerns I raised.

Please provide a more elaborate description of the experiments presented in Figure 6.

---

> ### Author Response · Authors · 2026-02-20
> **Clarification on requested comparison methods**
>
> We would like to express our sincere gratitude for your valuable and insightful feedback.
>
> We kindly request clarification regarding the comment that "a comparison with the other synthetic data generation methods mentioned in the Related Work section is missing". Note that to our knowledge this work is the first one that focuses on synthetic data generation from the point of view of instance-level recognition. Therefore, we would greatly appreciate it if the mentioned synthetic data generation methods could be specified. This will help us ensure that our revisions address the expectations accurately. In parallel, we are carefully working on other comments and conducting experiments. Once everything is completed, we will provide a comprehensive response addressing all points in detail.
>
> Thank you again for your time and patience.

---

> ### Author Response · Authors · 2026-04-27
> **Response to Reviewer Vjgr (1/2)**
>
> We thank the reviewer for the constructive feedback and for acknowledging the clarity of our work and the effectiveness of synthetic data. We address each concern below and highlight the revision in the manuscript in blue.
>
> - **Ambiguity in Instance Definition**. We added a dedicated study in **Supplementary Section B instance identity and diversity**, investigating whether the generated instances are sufficiently distinct and examining the effect of instance diversity. We included supporting evidence using five complementary approaches:
>   - **a. Qualitative visualization**. We visually inspected generated samples by selecting four object names in each of the four domains and generating several instances per object with different random seeds. Visual examples in Figures S1-S4 show that while certain instances may appear visually similar at first glance, they remain distinguishable through fine-grained details.
>   - **b. Human verification study**. To verify that instance identity is distinguishable and preserved after background and lighting variations, we conducted a human study in which participants had to identify the image originating from the same instance. The average accuracy of 96.25% demonstrates that, according to human annotators, generated instances are distinct and retain consistent identity after applying background variation.
>   - **c. Removing near-duplicates**. We applied the trained model (ID7) to compute similarities for all image pairs within each object, ranked pairs across all objects jointly, and removed one image from each of the top 500 most similar pairs that visually very similar. Note that 500 pairs is a tiny fraction of all 245k pairs (0.02%). Despite the reduced data, average performance improves slightly by 0.2 (see ID-S12 in Table S1), suggesting that near-duplicate instances have only an insignificant impact on overall performance.
>   - **d. Improving instance diversity**. We further increased diversity by enriching prompts with feature variations (see examples in Figure S7). Although the diversity of generations is improved visually, it does not yield improvement in performance (ID-S13 in Table S1), suggesting that the current amount of diversity is effective enough for good retrieval performance.
>   - **e. Failure in preserving instance identity**. We also presented failure cases in Figure S8 in which instance identity is not preserved well during generation. This is either due to the foreground object appearance getting modified for different backgrounds. Nevertheless, note that these cases are very rare cases that were hard to find.
>
> Overall, the results indicate that generated instances are sufficiently distinguishable and that removing near-duplicates and increasing diversity yield only marginal performance changes, demonstrating the robustness of our method.
> - **Reproducibility concerns**. We have added all implementation details of background generation in the main paper, including the specific tools used for background removal, the padding method, etc. We will release the complete pipeline, including object name lists, the generated dataset, and code upon publication to ensure full reproducibility.
> - **Lack of hyperparameter justification**. We have added an ablation study in **Supplementary Section E – Image generation hyperparameters**. We evaluate several combinations of the number of objects $C$ and instances per object $K$ while keeping the total number of training images ($C\times K$) constant. Results in Table S4 show that the best setting (ID-S23) surpasses the default setting (ID7) by only 0.1 on average. Overall results demonstrate our methodology is robust to a wide range of hyperparameter choices.

---

> ### Author Response · Authors · 2026-04-27
> **Response to Reviewer Vjgr (2/2)**
>
> - **Limited novelty and impact**.
>   - **Mix of real and synthetic data**. We appreciate this suggestion and have added experiments on combining real and synthetic data in **Section 4.2 – Training with both real and synthetic data**. Results shown in Table 5 indicate that progressively adding real data on top of synthetic data consistently improves performance. Overall, mixing real and synthetic data surpasses using either alone, indicating that synthetic data serves as an effective complement to real data, rather than only a substitute, for instance-level learning across various domains.
>   - **Comparison with other synthetic data generation methods**. To the best of our knowledge, no prior work has specifically explored synthetic data generation for universal instance-level image retrieval across diverse domains. Prior works start from an existing real training set and augment real images. In contrast, our setting assumes no real images and no instance annotations. The only input is a domain name, from which we synthesize instance-labeled training data and train a single retrieval model that generalizes to unseen real instances across domains. Our work is the first to investigate text-to-image generation for instance-level retrieval. The methods mentioned in Related work address different tasks or settings, making direct comparison not straightforward.
>   - **Novelty**. We respectfully disagree that the method is trivial. The main contribution is not the use of diffusion as an image editor, but the empirical finding that fully synthetic, text-prompted instance-level supervision is sufficient to improve foundation encoders for real-world instance-level retrieval. To emphasize the main empirical findings: synthetic-only training improves over pretrained encoders across diverse ILR benchmarks; it compares favorably to training on limited real instance-labeled data on average; and synthetic data is complementary to real data when both are available.
> - **Figure 6 description**. We apologize for the lack of clarity. Figure 6 compares retrieval performance between the pretrained model and our method (ID7). We have added clarification in the caption of Figure 6 and the description accordingly.
>
> We hope these revisions address the reviewer’s concerns and we are grateful for the feedback that meaningfully strengthened our work.

---

> > ### Comment · Reviewer_Vjgr · 2026-04-28
> > **Thanks.**
> >
> > Thank you to the authors for the detailed responses and revisions.
> >
> > I think my major concerns have mostly been resolved. The additional analyses and experiments, especially those on instance identity/diversity and the combination of real and synthetic data, strengthen the paper.

---

> > > ### Author Response · Authors · 2026-05-22
> > > **Thank you.**
> > >
> > > Thank you for the positive feedback. We are looking forward to hearing back from other reviewers too.

---

### Review · Reviewer_6bBr · 2026-01-28

**Summary Of Contributions:**

This work proposes a pipeline that augments a dataset by generating augmentations of an object with different backgrounds and/or lighting. The authors propose to use this pipeline to generate data for finetuning foundational image encoders to perform instance-wise metric learning for image retrieval.

**Audience:**

No

**Audience Explanation:**

Unfortunately, the idea of using some image generation pipeline to augment your training data is very obvious. Background removal as a first step is also obvious. One well cited paper is "Effective Data Augmentation With Diffusion Models" Trabucco et al. ICLR 2024. But there are many papers that incorporate this idea. Therefore, I do not see this result being of much use to the research community.

**Claims And Evidence:**

Yes

**Claims Explanation:**

The experiments are exhaustive. The ablation experiments are extensive. The authors verify their pipeline with a diverse set of benchmarks from various domains. They perform finetuning with different backbones and loss functions. They additionally perform extensive ablation studies on the image generation/augmentation process (Table 7).

The scale of the experiments is quite impressive, considering existing metric learning literature tends to focus on a standard set of old benchmarks (SOP, Cars, CUB/birds).

**Requested Changes:**

The submitted work could be useful as an open-source package that people can use to augment their dataset. Unfortunately, I do not see the findings in the submission to be useful otherwise.

One issue that stood out to me when reading the paper is that the proposed image augmentation pipeline is not able to change the orientation of the object. For example, looking at the page 1 figure, the augmentations of the object look like photoshopped versions of the original image. The object is simply scaled and translated, with some lighting changes. This limits the ability of the model to learn to associate different view angles of the same object, which is vital.

---

> ### Author Response · Authors · 2026-04-27
> **Response to Reviewer 6bBr**
>
> We thank the reviewer for the thorough evaluation and for acknowledging the exhaustive experiments, extensive ablations, and diverse set of benchmarks. We address the concerns below and highlight the revision in the manuscript in blue.
>
> - **Novelty of the pipeline**. We agree that diffusion-based data augmentation is now a well-studied direction. However, our setting differs substantially from conventional augmentation. Prior works such as Trabucco et al. start from an existing real training set and augment real images. In contrast, our setting assumes no real images and no instance annotations. The only input is a domain name, from which we synthesize instance-labeled training data and train a single retrieval model that generalizes to unseen real instances across domains. Our work is the first to investigate text-to-image generation for instance-level retrieval.
> The main contribution is therefore not the use of diffusion as an image editor, but the empirical finding that fully synthetic, text-prompted instance-level supervision is sufficient to improve foundation encoders for real-world instance-level retrieval. To emphasize the main empirical findings: synthetic-only training improves over pretrained encoders across diverse ILR benchmarks; it compares favorably to training on limited real instance-labeled data on average; and synthetic data is complementary to real data when both are available.
> - **Open source package**. We fully agree with the suggestion. We will release the complete pipeline, including all code and object name lists as an open-source package to enable the community to further explore the potential of synthetic data for instance-level representation learning. We will additionally release the generated image dataset used in our model training experiments.
> - **Viewpoint limitation**. While the original pipeline does not alter object orientation, we investigate whether incorporating viewpoint variation further enriches the generated data. We use TRELLIS [1], a cutting-edge image-to-3D model, to synthesize viewpoints from different angles (90 to 270 degrees). Details are added in **Supplementary Section E – Rendered viewpoints**. We sample four random views per instance before applying background generation. Examples shown in Figure S10 indicate a visually plausible rendering. However, results (ID-S31 in Table S5) indicate that the rendered viewpoints degrade performance on most datasets, likely due to 3D reconstruction artifacts, particularly for landmark objects. We include this analysis for completeness and believe that as image-to-3D models improve, incorporating multi-view augmentation will become more effective.
>   - **Real viewpoints**. The objaverse experiment presented in the original submission (now Table S6) additionally compares the case of a single 3D view to 4 different views. The benefit of 4 views is small even in this case of using real, and not synthetic, images.
>
> We hope these revisions address the reviewer’s concerns and we are grateful for the feedback that meaningfully strengthened our work.
>
> [1] Xiang et al. Structured 3D Latents for Scalable and Versatile 3D Generation. CVPR 2025.

---

> > ### Author Response · Authors · 2026-06-15
> > **Looking forward to additional feedback or comment**
> >
> > Thank you for engaging in the review process. We would be happy to receive any additional feedback or comments the reviewer may have.

---

### Review · Reviewer_X3Bq · 2026-04-14

**Summary Of Contributions:**

The paper addresses instance-level recognition (ILR) in a realistic setting where large-scale, exhaustively annotated instance-level training data is difficult to obtain. It proposes an automated pipeline that takes only a domain name or textual description as input, uses an LLM to generate object-category names, employs a text-to-image diffusion model to synthesize object instances, segments the foreground, and applies ICLight together with simple geometric transformations to create multiple background and appearance variations for each synthetic instance. These generated instance classes are then used to fine-tune foundation vision encoders with a retrieval-oriented metric learning objective.

The main empirical contribution is a comprehensive evaluation across seven ILR benchmarks covering artwork, landmarks, products, and multi-domain retrieval. The study includes comparisons with pretrained backbones, an Objaverse-based rendering baseline, matched-scale real-image training sets, multiple backbone architectures, several loss functions, and a range of ablations examining the number of positives per class, prompt design, LLM choice, diffusion model choice, and background generation strategy.

Overall, the paper presents a practically relevant and reasonably novel idea: improving ILR representations using fully synthetic instance-level training data, without relying on real instance photographs. Its main strengths are the breadth of the empirical study and the evidence that both realistic background generation and domain-targeted synthesis play an important role in the observed gains.

**Audience:**

Yes

**Audience Explanation:**

The paper should be of interest to parts of the TMLR audience, especially researchers working on representation learning, metric learning, image retrieval, synthetic data, and generative-model-based supervision. The problem is relevant and well motivated, since instance-level recognition is particularly expensive to supervise at scale, and the proposed domain-to-training-data pipeline is both practically useful and methodologically interesting.

The paper is also relevant beyond ILR itself. Its broader takeaway is that carefully constructed synthetic data, especially with controlled background and lighting variation, can improve learned retrieval representations. Because the evaluation spans seven benchmarks across multiple domains, the findings are likely to interest a wider audience than a narrowly focused single-domain retrieval study.

**Broader Impact Concerns:**

First, the method relies on pretrained LLMs and diffusion models to generate synthetic data in domains such as artworks, landmarks, and products. As a result, the generated images may still reflect copyrighted, trademarked, or otherwise restricted content present in the upstream training data. The paper should acknowledge these provenance and licensing limitations.

Second, the method may have dual-use implications. Improved instance-level retrieval can enable useful applications, but it could also support large-scale object tracking, counterfeit matching, or automated indexing of proprietary items.

Third, the generated dataset may inherit cultural, geographic, or stylistic biases from the LLM and diffusion model, which could affect coverage and contribute to uneven performance across domains.

**Claims And Evidence:**

Yes

**Claims Explanation:**

The claims are mostly supported, but not all of the stronger conclusions are fully justified. The experiments provide solid evidence that the proposed synthetic instance-level data can improve pretrained retrieval models, and the comparisons with pretrained backbones, real-data baselines, and ablations make this central result credible.

However, the evidence is less convincing for broader claims about generality. The gains are not uniform across datasets, real-data training is still stronger in some domain-specific cases, and the paper does not report variance, repeated runs, or significance analysis. In addition, a key assumption—that different random seeds correspond to distinct instances—is not directly validated. The train/test leakage check is also helpful but limited.

Overall, the paper supports the claim that the proposed pipeline is useful and often effective, but stronger validation is needed for its broader claims.

**Requested Changes:**

1-  Provide a more direct validation of the main instance-generation assumption. The method relies on the assumption that different random seeds produce distinct object instances within the same category, while background and lighting variations preserve the identity of each instance. Since this is central to the approach, it should be validated more explicitly, for example through human evaluation, pairwise identity verification, or a quantitative analysis of intra-class consistency and inter-class separability on sampled generated data.

2-  Report robustness across multiple runs. The results would be more convincing if the paper included variance across several training runs and, ideally, across different generation seeds as well. Because some of the reported gains are relatively modest, confidence intervals or statistical significance tests would help establish that the improvements are reliable.

3-  Strengthen the analysis of failure cases relative to real-data training. The comparison in Table 4 already shows that real training data remains superior in some in-domain settings, particularly for landmarks. This limitation deserves a more detailed discussion, ideally supported by qualitative examples and an analysis of the kinds of object properties or structures that are difficult for the generative pipeline to model accurately.

4-  Make the train/test leakage analysis more rigorous. The current visual inspection of the most similar retrieved pairs is a useful preliminary check, but it is not sufficient for a method that depends heavily on generative models. A stronger analysis could include nearest-neighbor overlap statistics, duplicate or near-duplicate detection, or retrieval using stronger external descriptors to more systematically rule out memorization or unintended overlap.

5-  Move more reproducibility details from the supplement into the main paper. In particular, the paper would benefit from clearer reporting of prompts, filtering decisions, segmentation settings, and any generation failures or rejection criteria. The method is generally understandable, but these implementation details are important for reproducibility and practical reuse.

6- Include a comparison against a mixed-data regime, such as synthetic-only vs. few-shot real + synthetic. That would help readers understand whether the method is mainly a substitute for real data or a complement to it.

7- Add more qualitative failure cases. The examples and ablations are helpful, but the paper would be stronger with explicit examples where background synthesis harms identity preservation, where segmentation fails, or where generated instances are not sufficiently distinct.

---

> ### Author Response · Authors · 2026-04-27
> **Response to Reviewer X3Bq (1/2)**
>
> We thank the reviewer for the thorough feedback and for recognizing the novelty and comprehensiveness of our study. We address each point below and highlight the revision in the manuscript in blue.
>
> **1. Validation of the instance-generation assumption**. We added a dedicated study in **Supplementary Section B instance identity and diversity**, investigating whether the generated instances are sufficiently distinct and examining the effect of instance diversity. We included supporting evidence using five complementary approaches:
>   - **a. Qualitative visualization**. We visually inspected generated samples by selecting four object names in each of the four domains and generating several instances per object with different random seeds. Visual examples in Figures S1-S4 show that while certain instances may appear visually similar at first glance, they remain distinguishable through fine-grained details.
>   - **b. Human verification study**. To verify that instance identity is distinguishable and preserved after background and lighting variations, we conducted a human study in which participants had to identify the image originating from the same instance. The average accuracy of 96.25% demonstrates that, according to human annotators, generated instances are distinct and retain consistent identity after applying background variation.
>   - **c. Removing near-duplicates**. We applied the trained model (ID7) to compute similarities for all image pairs within each object, ranked pairs across all objects jointly, and removed one image from each of the top 500 most similar pairs that visually very similar. Note that 500 pairs is a tiny fraction of all 245k pairs (0.02%). Despite the reduced data, average performance improves slightly by 0.2 (see ID-S12 in Table S1), suggesting that near-duplicate instances have only an insignificant impact on overall performance.
>   - **d. Improving instance diversity**. We further increased diversity by enriching prompts with feature variations (see examples in Figure S7). Although the diversity of generations is improved visually, it does not yield improvement in performance (ID-S13 in Table S1), suggesting that the current amount of diversity is effective enough for good retrieval performance.
>   - **e. Failure in preserving instance identity**. We also presented failure cases in Figure S8 in which instance identity is not preserved well during generation. This is either due to the foreground object appearance getting modified for different backgrounds. Nevertheless, note that these cases are very rare cases that were hard to find.
>
> Overall, the results indicate that generated instances are sufficiently distinguishable and that removing near-duplicates and increasing diversity yield only marginal performance changes, demonstrating the robustness of our method.
>
> **2. Multiple runs**. We now report the mean and standard deviation over three training seeds in Table 3. The deviation is noticeably smaller than the difference across different ILGen variants.
>
> **3. Failure cases**. The revision presents and comments on the following failure cases.
>   - a. Failure of preserving instance identity as discussed in 1e above. Comparing this problem of generated data to real image training data: This issue is unique to generated data and is not aligned with real-world images. We observe it more in the landmarks domain which may be justifying the lower performance of ILGen compared to real data in Table 4.
>    - b. Generating near duplicate objects as discussed in 1c above. Comparing this problem of generated data to real image training data: ILR labeling of real images is tough to obtain at a very large scale, having duplicates is likely due to the scale challenge and annotation errors. Actually, we have observed such cases in the GLDv2 training set.
>
> **4. Train/test leakage analysis**. To expand train-test leakage analysis, we additionally applied the copy detection model SSCD to compute similarity between generated training and real test images across all datasets (see **Section 4.3 - Train and test set overlap**). We present the top similar pairs between the train and each test dataset in Figure 8. The highest similarity score 0.38 is strictly below the copy-detection threshold of 0.5 following [1], indicating that none of the generated images are copies of the test images. Manual inspection of the most similar pairs confirms this.
>
> **5. Reproducibility details in the main paper**. We moved key details from the supplementary into the main paper, including the background generation details, training details, and losses used in the ablation study.

---

> ### Author Response · Authors · 2026-04-27
> **Response to Reviewer X3Bq (2/2)**
>
> **6. Mixed-data regime**. We appreciate this suggestion and have added experiments on combining real and synthetic data in **Section 4.2 – Training with both real and synthetic data**. Results shown in Table 5 indicate that progressively adding real data on top of synthetic data consistently improves performance. Overall, mixing real and synthetic data surpasses using either alone, indicating that synthetic data serves as an effective complement to real data, rather than only a substitute, for instance-level learning across various domains.
>
> **7. Ethical considerations**. We appreciate the suggestions on stating potential ethical concerns. We added **Section A in the supplementary** and discussed provenance and licensing concerns, dual-use risks, and inherited biases from generative models.
>
> We hope these revisions address the reviewer’s concerns and we are grateful for the feedback that meaningfully strengthened our work.
>
> [1] Somepalli et. al. Diffusion art or digital forgery? Investigating data replication in diffusion models. CVPR 2023.

---

> > ### Comment · Reviewer_X3Bq · 2026-05-27
> > **Recommendation**
> >
> > Thank you to the authors for clarifying my concerns. Most of my concerns have been adequately addressed, and therefore I recommend acceptance. I strongly encourage the authors to incorporate all the proposed modifications in the final submission.

---

### Decision · Action_Editor_AayY · 2026-06-22

**Recommendation:** Accept as is

**Audience:**

Yes

**Audience Explanation:**

Reviewer 6bBr initially answered "no" to this question, stating that some of the ideas are already known.  The authors responded well to this point, and the reviewer ultimately said they were happy with the response, and voted to recommend acceptance.

The other reviewers were positive on the question of whether there would be some people interested in the findings of the paper.  Certainly, the topic of the paper is broadly of interest to many in the community.

**Claims And Evidence:**

Yes

**Claims Explanation:**

Two of the reviewers initially answered "no" to this question (X3B1 and Vjgr), with various concerns about the experimental results.  The authors responded by adding in a new section into the supplementary material (Supplementary Section B) that addressed each of these concerns fairly exhaustively.  As a result, both reviewers indicated that they were happy with the changes made (with only a comment to make sure that these changes are included in the final version).  Both reviewers indicated that they are in favor of the paper being accepted.

The other reviewer was largely positive on this question in their initial review, and their criticisms were also addressed during the rebuttal period.